# Imaging and SERS Study of the Au Nanoparticles Interaction with HPV and Carcinogenic Cervical Tissues

**DOI:** 10.3390/molecules26123758

**Published:** 2021-06-20

**Authors:** Andrea Ceja-Fdez, Ramon Carriles, Ana Lilia González-Yebra, Juan Vivero-Escoto, Elder de la Rosa, Tzarara López-Luke

**Affiliations:** 1Departamento de Física Médica, División de Ciencias e Ingenierías Campus León, Universidad de Guanajuato, León 37150, Mexico; a.ceja@ugto.mx; 2Centro de Investigaciones en Óptica, A.P. 1-948, León 37150, Mexico; ramon@cio.mx; 3Departamento de Ciencias Aplicadas al Trabajo, División Ciencias de la Salud, Campus León, Universidad de Guanajuato, León 37670, Mexico; analilia@ugto.mx; 4Department of Chemistry, The University of North Carolina at Charlotte, 9201 University City Blvd., Charlotte, NC 28223, USA; juan.vivero-escoto@uncc.edu; 5Facultad de Ingenierías, Campus Campestre, Universidad De La Salle Bajio, León 37150, Mexico; edelarosa@delasalle.edu.mx; 6Instituto de Investigación en Metalurgia y Materiales, Universidad Michoacana de San Nicolás de Hidalgo, Edificio U, Ciudad Universitaria, Morelia 58030, Mexico

**Keywords:** gold nanoparticles, HPV, cervical cancer, two-photon imaging, confocal microscopy, multiphoton microscopy, Raman spectroscopy and SERS

## Abstract

In this work, gold NPs were prepared by the Turkevich method, and their interaction with HPV and cancerous cervical tissues were studied by scanning electron microscopy, energy-dispersive x-ray spectroscopy, confocal and multiphoton microscopy and SERS. The SEM images confirmed the presence and localization of the gold NPs inside of the two kinds of tissues. The light absorption of the gold NPs was at 520 nm. However, it was possible to obtain two-photon imaging (red emission region) of the gold NPs inside of the tissue, exciting the samples at 900 nm, observing the morphology of the tissues. The infrared absorption was probably due to the aggregation of gold NPs inside the tissues. Therefore, through the interaction of gold nanoparticles with the HPV and cancerous cervical tissues, a surface enhanced Raman spectroscopy (SERS) was obtained. As preliminary studies, having an average of 1000 Raman spectra per tissue, SERS signals showed changes between the HPV-infected and the carcinogenic tissues; these spectral signatures occurred mainly in the DNA bands, potentially offering a tool for the rapid screening of cancer.

## 1. Introduction

According to the World Health Organization, cervical cancer was the fourth most frequent cancer in women, with an estimated 530,000 new cases in 2012. In the same year, this disease was responsible for approximately 7.5% of all female cancer deaths worldwide. The mortality associated with cervical cancer could be significantly diminished if it were detected at the early stages of evolution or at the precancerous stage, denominated cervical intraepithelial neoplasia (CIN) [1]. It is well established that a long-term infection with human papillomavirus (HPV), especially types 16 and 18, is a major cause for the development of cervical pre-cancer and cancer [2,3]. The Papanicolaou (Pap) smear test is the most widely used method to detect cervical tissue damage; unfortunately, while this test has good specificity, as high as 95–98%, it also has a low sensitivity of approximately 50% [4]. Techniques such as automated cytology and HPV testing have been used to reduce the false negative results [5,6,7]. When a Pap smear result is abnormal, the clinical diagnosis is complemented by colposcopy, biopsy and visual examination of the histological sections; however, the grading scale used by health professionals is subjective, due to human intervention, and sometimes pre-malignancy may not even be visible at all.

The search for new diagnostic tools with better specificity and sensitivity has led to the consideration of different spectroscopic techniques capable of detection at the molecular level. Over the past years, Raman spectroscopy has provided excellent results in the diagnosis of a wide range of cancers, including breast, prostate, esophageal, colon, lung, oral and cervical [8,9,10,11,12,13,14]. Raman spectroscopy can be implemented microscopically to study cellular components such as nucleus, nucleoli, or chromatin [10,15]; due to its high spatial resolution, specific biomolecular complexes such as phenylalanine, tyrosine, and adenine can be analyzed [16,17,18]. The Raman effect can be greatly enhanced using Surfaced Enhanced Raman Scattering (SERS); this technique can detect analytes down to the single molecule level retaining molecular specificity [19,20,21]. The sensitivity of SERS spectroscopy can be dramatically improved by using either gold or silver nanostructures to obtain enhancement factors of several orders of magnitude, due to the local field effect induced by the surface plasmon resonance from the nanostructured surfaces [16,22,23,24]. Nevertheless, the local field enhancement is effective only when Raman tags are closely located (usually less than 100 nm) to the substrate surface.

One advantage of gold nanoparticles (Au NPs) for biomedical applications is that the plasmon resonance frequency can be tuned through control of the nanoparticle size, morphology, or geometry. In particular, it is possible to tune the gold nanoparticles absorption wavelength to the near-infrared region not absorbed by the tissue, allowing a true diagnosis [25,26,27]. In addition, it is possible to take advantage of the field enhancement effect from Au NPs with other techniques for example, with two-photon imaging (TPI). This powerful microscopic image acquisition tool is non-invasive and uses near-infrared excitation, leading to large penetration depths of up to hundreds of microns [28,29,30]. It has been reported that TPI can be observed from roughened surfaces due to the resonant coupling of specific light frequencies to the surface plasmon resonances (SPR) that lead to local electric field enhancement [28,31,32]. Traditionally, samples for TPI have been marked with organic fluorophores [33] but recently other contrast agents have been tried, such as quantum dots [34], metallic nanoparticles [35,36,37,38], and gold nanorods [39,40].

In this work, we evaluated the interaction of gold nanoparticles with HPV and cervical cancer tissues. Microscopic images were acquired using SEM, chemical mapping, confocal and two-photon imaging of the samples to demonstrate the introduction of gold nanoparticles to the tissues. Two-photon imaging (red region) excited at 900 nm was obtained and a better image contrast was observed. We observed the Au NPs agglomerated in the tissues, promoting the local field effect, resulting in a favorable effect for SERS technique which is important to identify enhanced bands that correspond to different pathologies giving the opportunity of differentiate them. SERS signal of DNA and amide III bands in labeled cancer tissue are until 10 times stronger than labeled VPH tissue. This result could made possible to differentiate the presence of cervical cancer at early stage. It is the result of the plasmon oscillation confinement estimated at 10^4^ Au NPs interacting with such molecules and highlights the relevance for cervical cancer detection. These preliminary results allowed identification of differences between HPV and cancer cervical tissues, with the Au NPs offering possibilities for future alternative as markers and sensors.

## 2. Materials and Methods

### 2.1. Synthesis of Gold Nanoparticles 

The Turkevich method was used to synthesize the gold NPs with some modifications [41]. Briefly, 250 µL of a 0.1 M HAuCl_4_ solution (prepared by dissolving 0.5 g of HAuCl_4_ (99.999%, Sigma-Aldrich, Toluca, México) in 14.715 mL of deionized water) and 3.75 mL of 1% wt. sodium citrate solution (obtained by dissolving 0.0375 g of Na_3_C_6_H_5_O_7_ (Sigma-Aldrich) in 3.75 mL of deionized water) were added into 25 mL of boiling water under stirring (400 rpm). After a few minutes the solution turned ruby red, at this time the solution was cooled down to room temperature and the resulting colloid was filtered. The final product was kept in storage at 4 °C.

### 2.2. Gold Nanoparticles in Cervix Tissue

Formalin-fixed paraffin preserved (FFPP) cervical tissue samples were obtained by an experienced breast pathologist from the Pathology Department of the Instituto de Seguridad y Servicios Sociales de los Trabajadores del Estado (ISSSTE) in Guanajuato State (Mexico). A total of 15 samples with different grades of CIN were identified. Of those samples, 5 presented low-grade squamous intraepithelial lesions (CIN1); other five showed medium grade squamous intraepithelial lesions (CIN2), and the final five presented high-grade squamous intraepithelial lesion (CIN3). These cases were randomly selected; each case represents a sample from an individual patient. For evaluation by the pathologist, the samples were dewaxed by immersion in baths of xylene (KARAL) and absolute ethanol (KEM). For the Au NPs study, two parallel 5 μm-thick FFPP sections were cut from each block using a microtome, mounted on glass slides and dried.

To prepare the cervix tissue for impregnation with the Au NPs, the histological specimens embedded in paraffin and mounted on glass microscope slides were heated to 60 °C during 1 h. To deparaffinize the mounted tissues, they were immersed in xylene twice (15 min each); then the tissues were rehydrated by using different concentrations of absolute EtOH as follows: EtOH 100% (10 min), EtOH 90% (10 min), EtOH 70% (10 min) and EtOH 30% (10 min). Finally, the samples were immersed in PBS 1X (5 min) and rinsed in distilled water (5 min).

To stain the deparaffinized tissues, the water excess was removed, and the specimens outline was marked with a hydrophobic ink (Pap-pen^®^), avoiding making contact with the tissue, this was made to maintain the colloidal solution concentrated only in the tissue area. The specimens were placed on a hot-bar and incubated with the Au NPs colloidal solution for 2 h at 37 °C. The tissue dryness was continuously checked; if required, more gold solution was added. Afterwards, the tissue was carefully rinsed with distilled water. For characterization using confocal microscopy, a few drops of DABCO (1,4-diazabicyclo[2.2.2]octane) were added. Finally, the samples were covered with a coverslip making sure that no bubbles were visible. Following a protocol used to preserve oral cells for chemical analysis by Raman spectroscopy [13], DABCO was not added to the samples studied by Raman micro-spectroscopy; instead, these tissues were dried at 90 °C for 3 min.

### 2.3. Characterization

The morphology and size of the Au NPs in water solution were analyzed by transmission electron microscopy (TEM) using a FEI Titan 80–300 electron microscope (Hillsboro, OR, USA) with an accelerating voltage of 300 kV. Tissue was analyzed by a Scanning Electron Microscope (SEM) JEOL JSM-7800F (Tokyo, Japan) and by a SEM-Energy Dispersive X-ray Spectrometer (EDS) from Oxford Instruments (Abingdon, UK). UV-Vis absorption measurements were carried out using a Perkin Elmer Lamda 900 spectrometer (Waltham, MA, USA) with a spectral resolution of 2 nm. Optical characterization of cervix tissue was obtained by a Zeiss confocal microscope, model LSM-710-NLO, equipped with a Chameleon Vision-II ultrafast laser system from Coherent (tunable from 680 to 1040 nm, 140 fs, 80 MHz repetition rate). The excitation wavelength used for confocal images was 543 nm and 900 nm for TPI.

The SERS spectra were acquired using an inVia Raman microscope from Renishaw with a 20×/0.4 objective and approximately 5 mW of excitation power at 785 nm wavelength. The integration time for each Raman spectrum was 20 s and the spectral range was from 700 to 1700 cm^−1^. The Raman data was obtained by mapping a minimum of 10 independent tissue regions in each sample, depending on the quality of each slide, each site was measured twice. The results represent the average of all the mapped regions for each sample.

## 3. Results and Discussion

The size and morphology of the Au NPs were studied by TEM; a typical micrograph of the particles is presented in Figure 1a, clearly, the shape of the nanoparticles is an elongated circle. The UV-VIS absorption spectra of Au NPs dispersed in aqueous solution is shown in Figure 1b. In all cases, the SPR was centered at 520 nm which is consistent with the particle size and the observed ruby-red color of the Au NPs obtained with the Turkevich method [42]. A lower magnification micrograph is shown in Figure 1c; the inset presents a histogram of nanoparticle size distribution; the average size is approximately 13 ± 2 nm.

In order to evaluate the presence of the NPs at the cellular level, we acquired SEM micrographs of the carcinogenic cervix cells (Figure 2a,b) and cells with HPV presence (Figure 2c,d). The cell size differences between cancerous (Figure 2a) and HPV-infected (Figure 2c) cells are very clear; approximately three times bigger the carcinogenic one than the HPV cell, this cell enlargement is common when a damage is present [43]. Panels (b) and (d) in Figure 2 show the EDS maps of gold distribution in the same sample region, cancerous and HPV, respectively. The distribution of Au NPs was observed within the cells. Our hypothesis to explain this behavior is that the Au NPs have negative charge in their surface while the nucleus could be positively charged [44].

Once we observed the location and chemical identity of the Au NPs, we acquired optical microphotographs using confocal and two-photon microscopy. All images were taken with an epi-plan 50×/0.55 objective. Before acquiring the TPI images, we systematically tuned the laser wavelength from 800 to 1100 nm, in 50 nm increments, to determine the optimal excitation wavelength. We found that 900 nm yielded the brightest images; this wavelength corresponds to the frequency of the plasmon resonance of the nanoparticles.

Figure 3 displays some characteristic microscopic images from cancerous cervical tissue with and without Au NPs. The first column corresponds to confocal images, the second to two-photon imaging (TPI), and the third presents compound images superimposing the corresponding panels from the previous columns. Panels (a–c) present images from cancerous tissue incubated with Au NPs; panels (d–f) show a close-up view of the red circled area in panel (a); finally, panels (g–i) correspond to cancerous tissue with no Au NPs. Comparing Figure 3g with the panels on the first row, we observe that the presence of Au NPs has two main consequences in the autofluorescence signal from the cells, the signal from the cell nucleus quenches and the cell boundaries are blurred. The cells without NPs have relatively well-defined boundaries but after incorporation of the NPs the boundaries were no longer distinguishable in the confocal image. On the other hand, when we compared the second column in Figure 3, the TPI signal obtained at red emission region from the cell nuclei increased substantially after the incorporation of the Au NPs, as expected. The red shift emission of the Au NPs is possible due to the NPs aggregation inside of the tissue. On the contrary, the TPI signal from the cytoplasm, which does not have a significant concentration of Au NPs, is significantly weaker than the signal from the nucleus. The TPI signal from the nucleus is consistent with the observation of gold location from EDS. Thus, we can conclude that Au NPs could be used as marker for cell nucleus.

It has been reported that cervical tissue shows SERS peaks around 722, 755, 782, 849, 853, 921, 938 and 1245 cm^−1^ [17,45,46]. Figure 4 shows three different micro-SERS spectra of our cervical tissue samples excited at 785 nm; the top panel corresponds to HPV-infected tissue marked with Au NPs, the bottom spectra are from tissue with carcinoma, where the top line corresponds to tissue marked with Au NPs and the bottom line to unmarked tissue. The Raman spectrum signal level of the unmarked tissue is very weak, while the signal strength from the marked tissues is strong with well-defined peaks. The most prominent peaks we observed in HPV and carcinogenic tissues with Au NPs are listed in Table 1 [10,17,47].

In the comparison of SERS spectra between cervical cancer tissue with respect to the HPV ones several differences in spectral intensities are reveled, principally in regions located in the ranges of 920–1140 cm^−1^, 1160–1280 cm^−1^ and 1290–1480 cm^−1^, respectively. Figure 5 shows a plot of the integrated signal of the 920–1140 cm^−1^, 1160–1280 cm^−1^ and 1290–1480 cm^−1^ regions of vibrational bands, Figure 5A shows the integrated signals of HPV tissue and Figure 5B the carcinogenic tissue ones. Each data point represents the average value from three SERS regions and error bars show the standard deviations, see Appendix A. According with these results it could be observed that variations in carcinogenic SERS spectra are lower than HPV ones, particularly in the region of 920–1140 cm^−1^ that mostly corresponds to DNA bands; this can be due to the deformations involved in cell damage caused by HPV infection. Table 1 lists tentative assignments for the observed SERS bands, according to the literature.

Comparing the signals from HPV-infected tissue (Figure 4, top panel) and invasive carcinoma (Figure 4, lower panel), we observed the following spectral changes: (i) modification of lines associated with saccharides at 849, 889 and 917 cm^−1^ due to glycogen skeletal deformation, CCH aromatic deformation and CCH deformation, respectively [48], marked by red arrows in Figure 4; (ii) collagen bands at 1277 and 1401 cm^−1^ (green arrows) and CH_2_CH_3_ deformation at 1456 cm^−1^ (black arrow) are present in the HPV-infected tissue but disappear in the cancerous samples. These results corroborate previous reports stating the principal differences between normal cervical tissue and invasive cervical-cancer tissue are primarily on the basis of collagen bands and CH_2_ stretching bands [44]. Cervical cells are unusual among other epithelial cells in that they accumulate large amounts of glycogen during the maturation process [48]. Glycogen is known to be linked to cellular maturation and disappears with the loss of differentiation during neoplasia.

Beside the aforementioned lines, invasive carcinoma also shows characteristic nucleic acid bands such as 970, 1084, 1181 and 1373 cm^−1^ [17,45,46,47,48]. An increment in the intensity of the amide III bands at 1228 and 1254 cm^−1^, and the amide II band at 1558 cm^−1^ was observed in the spectra of carcinoma samples as compared to the HPV tissue samples, see magenta arrows in Figure 4. The increment in nucleic acids and proteins result from an increased proliferation of these substances in the tumor cells. In particular, the 1228 cm^−1^ band associated with amide III of beta sheeted proteins could be used to differentiate normal and early stage cancerous cervix tissues. It has been reported that bands associated with high-grade squamous intraepithelial lesion tissue (HSIL)—such as 738 and 1373 cm^−1^ (thymine), 785 cm^−1^ (cytosine) or 890 cm^−1^ (saccharides)—are associated with DNA; therefore, they indicate the presence of rapidly proliferating cells having high DNA content [45].

Some researchers have suggested that Raman spectra associated with the CIN2 lesion cluster with the signal of the basal epithelial cells of the normal squamous epithelium; therefore, this signal shares indistinctive biochemical profiles, such as the bands that are associated with the amide I and amide III bands [49]. In our case, the Raman spectra of carcinogenic tissues show an increased contribution from amide III and phosphate stretching, at 1252 cm^−1^ indicated by the second magenta arrow in Figure 4, and contributions from guanine, thymine and C–H deformation in proteins and carbohydrates at 1342 and 1373 cm^−1^ as shown by the blue arrows in the 1300 cm^−1^ region. In this work, we propose that the increased presence of Au NPs in the nuclei of the cervix tissue cells leads to useful SERS spectra for possible aid in early cancer diagnosis. The agglomeration of Au NPs generates “hot-spots” which increase the signal of the molecules close to the nanoparticles. The SERS technique was important to identify enhanced bands that correspond to different pathologies giving the opportunity of differentiating them.

In order to elucidate the enhancement factor (EF) by the introduction of Au NPs, we analyze three characteristic Raman bands showing strong change between infected and cancer tissue, 1084 cm^−1^ (nuclei acid), 1228 and 1254 cm^−1^ (amide III) and 1300–1450 cm^−1^ (CH, thymine, guanine, adenine, collagen and CH_2_), see Figure 4. Interestingly, those bands show an EF of Raman signal of ~10^3^ for Au NPs labeled compared to unlabeled cancer tissue, after the contribution of the VPH-infected tissue is subtracted. Furthermore, Raman signal of such bands in labeled cancer tissue are 6, 8 and 10 times stronger than labeled VPH tissue, respectively. This large difference made possible to elucidate the presence of cervical cancer at early stage. It is the result of the plasmon oscillation confinement of Au NPs interacting with such molecules and highlights the relevance for cervical cancer detection. For this calculation, it is estimated ~10^4^ Au NPs inside the laser spot size area of ~3.02 μm^2^.

## 4. Conclusions

In this study, the microstructural and SERS spectroscopic differences between HPV-infected and carcinogenic tissues embedded with Au NPs were evaluated. SEM images showed changes in cell nuclei sizes of up to 300%, which is common when cellular damage occurs. EDS corroborated that Au NPs were located mostly in the cell nucleus. We hypothesize that this is due to the charge differences, Au NPs being negatively charged and the nucleus being positive. Two-photon imaging of the morphology of the tissue embedded in AuNPs was obtained in the red region, excited at 900 nm. SERS spectroscopy, enabled by the local field enhancement effect produced by the Au NPs, revealed several changes between HPV-infected and carcinogenic tissues mainly in bands corresponding to DNA markers. A spectroscopic approach that yields compositional information of cell nuclei could be a powerful tool for rapid cell characterization and assessment of cellular activities at the sub-cellular level.

## Figures and Tables

**Figure 1 molecules-26-03758-f001:**
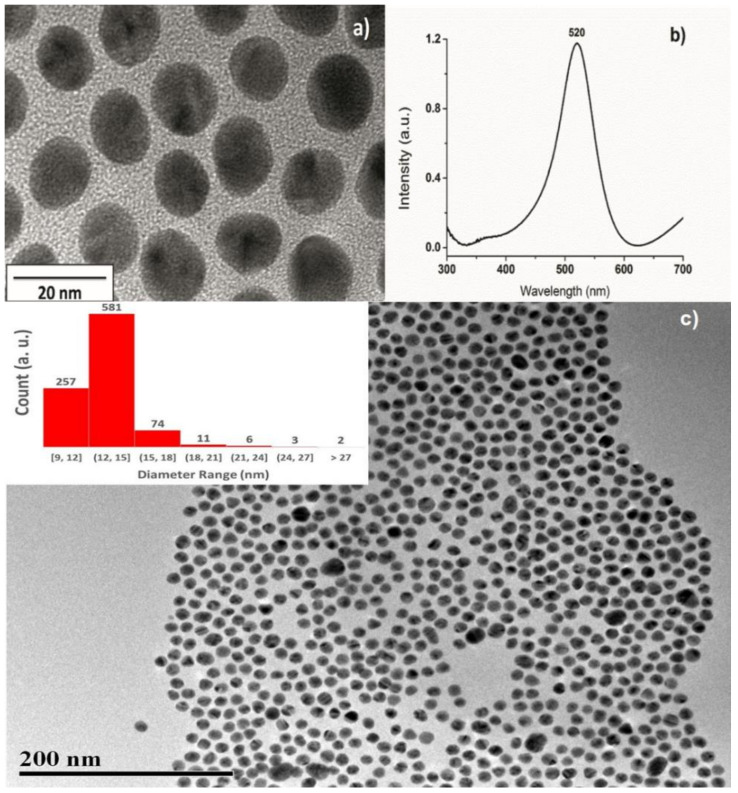
(**a**) High magnification TEM micrographs of gold nanoparticles; (**b**) absorption spectra of the gold nanoparticles; (**c**) TEM micrograph, inset shows the size distribution of gold nanoparticles.

**Figure 2 molecules-26-03758-f002:**
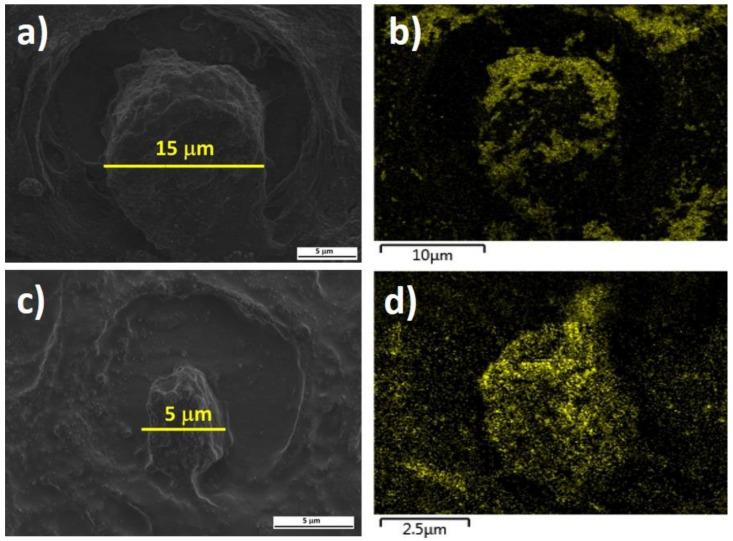
SEM and EDS micrographs of cancerous and HPV-infected tissues impregnated with Au NPs (**a**) SEM of carcinogenic cell; (**b**) EDS map of gold (shown in green) in the same region as panel (**a**); (**c**) SEM of HPV-infected cell; (**d**) EDS map of gold in the same region as panel (**c**). Note the difference in scale bars for each panel.

**Figure 3 molecules-26-03758-f003:**
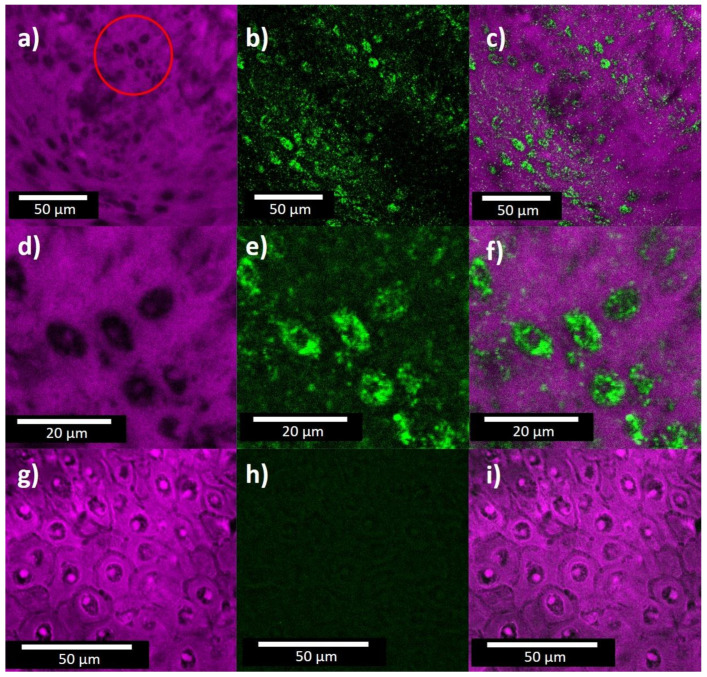
Micrographs of carcinogenic cervical tissue, (**a**) confocal image of tissue with Au NPs, excited at 543 nm; (**b**) two-photon imaging of cervical tissue with Au NPs, excited at 900 nm; (**c**) superposition of images from panels (**a** and **b**); (**d**–**f**) zoom in of the red circled region in panel (**a**); (**g**–**i**) confocal, TPI and superposition of both images, respectively, of cervical tissue without Au NPs.

**Figure 4 molecules-26-03758-f004:**
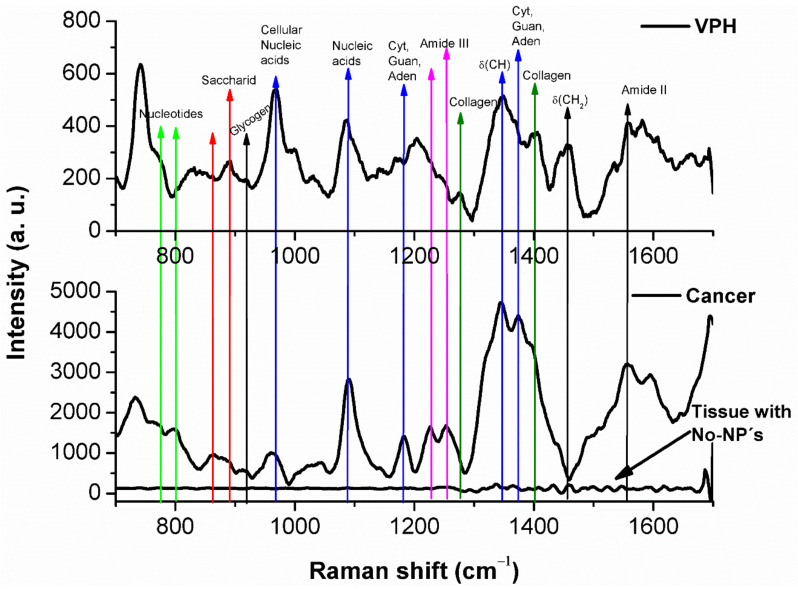
SERS spectra of cervical tissues. **Top panel**, tissue infected with HPV and marked with Au NPs; **bottom panel**, tissues with carcinoma, one marked with Au NPs the other without NPs.

**Figure 5 molecules-26-03758-f005:**
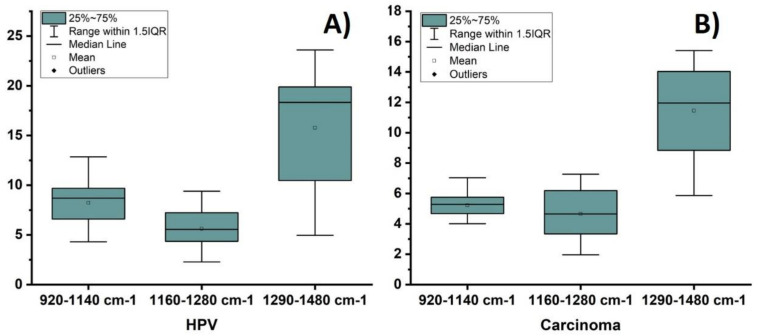
A plot of the integrated SERS regions 920–1140 cm^−1^, 1160–1280 cm^−1^ and 1290–1480 cm^−1^, respectively. Each point represents the average value from three SERS spectra and error bars show the standard deviations. SERS spectra of cervical tissues. (**A**) Tissue infected with HPV and marked with Au NPs; (**B**) tissues with carcinoma marked with Au NPs.

**Table 1 molecules-26-03758-t001:** Peaks assignments for SERS spectra of HPV and carcinogenic tissues with Au NPs. [17,44,45].

HPV Tissue		Carcinogenic Tissue	
Peak (cm⁻¹)	Assignment	Peak (cm⁻¹)	Assignment
741 774 890 919 967 1002 1030 1087 1167 1203 1277 1345 1400 1456 1554 1665	DNA DNA Proteins Glycogen Proline, Valine Phenylalanine Phenylalanine DNA Proteins Tryptophan Amide III DNA Collagen Lipids Amide II Amide I	734 774 864 919 967 1228 1252 1345 1373 1400 1554 1595 1665	DNA DNA Hidroxiproline Glycogen Proline, Valine Phenylalanine Amide III DNA Thymine Collagen Amide II Lipids Amide I

## Data Availability

Data is contained within the article or Appendix A.

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
