# Peer review of "Imaging and SERS Study of the Au Nanoparticles Interaction with HPV and Carcinogenic Cervical Tissues"

_molecules, 2021, doi:10.3390/molecules26123758_

Round 1
Reviewer 1 Report
In the submitted manuscript, Dr. López-Luke et al. investigated the interaction between citrate stabilized AuNPs and HPV or Carcinogenic Cervical Tissues. The manuscript is well written, and the results are well explained. However, the aim of this work is not clear.
The authors have to give answers to the next questions:
- It is known that the SERS is a time-consuming technique and requires a diagnostic laboratory with well-skilled staffing. Therefore, if the authors observed significant differences in dimensions at the SEM micrographs of these two different tissues, why they used SERS? Why is this measurement important? Is it additional confirmation of HPV infected tissue or carcinogenic tissues? Please justify the purpose of SERS measurements deeply?
- Could the authors provide the micrographs of HPV infected tissue and compare them with the micrographs of carcinogenic cervical tissue? Are the same or exist differences between them?
Please correct reference 1. I observed the error in it.
In the end, in this form, the manuscript satisfies the standards of this journal, but I have to suggest a minor revision before acceptance.
Reviewer 2 Report
The paper deals with a SERS study of gold nanoparticles attached to human papillomavirus (HPV) and carcinogenic cervical tissues. The authors have used the classical strategy for making gold nanoparticles by the citrate method. HPV infected tissues were impregnated with the resulting citrate-stabilized spherical gold nanoparticles for chemical analysis by Raman spectroscopy. Surprisingly, the brightest images were found at 900 nm excitation wavelength in the biological window, suggesting a gold nanoparticle aggregation at the tissue surface. The reference list is incorrect (Ref. 39 is not Turkevich!) and not inherent.
Points of criticism:
- Introduction:
Relevant references about asymmetric gold nanoparticles (e.g. nanotriangles in Colloids Surfaces B 167 (2018) 560-567 or nanostars in RSC Advances 9 (2019) 23633-23641) with UV-vis absorption maxima tuned in in the first and second window for in vivo imaging are missing.
- Results:
Figure 2:
The scale bars in Figure 2 should be identical in all micrographs! It is not clear how it is possible to realize SEM micrographs in absence and presence of AuNPs exact at the same point?
A nanoparticle aggregation (accompanied by a shift of the plasmon resonance maximum to 900 nm as already discussed in the text) should be visualized in the SEM micrographs at higher magnification!
Figure 4:
The plasmon resonance enhancement factor EF should be verified and discussed in relation to the relevant literature.
- The manuscript should be carefully checked again:
Figures or figures; Fig or Fig. ? on page 7
- The reference list has to be checked carefully again!

Round 2
Reviewer 2 Report
The paper is now acceptable.